# Combining Cluster Analysis of Air Pollution and Meteorological Data with Receptor Model Results for Ambient PM_2.5_ and PM_10_

**DOI:** 10.3390/ijerph17228455

**Published:** 2020-11-15

**Authors:** Héctor Jorquera, Ana María Villalobos

**Affiliations:** 1Departamento de Ingeniería Química y Bioprocesos, Pontificia Universidad Católica de Chile, Santiago 7820244, Chile; 2Centro de Desarrollo Urbano Sustentable, Pontificia Universidad Católica de Chile, Santiago 7520245, Chile; 3DICTUC S.A., Vicuña Mackenna 4860, Santiago 7820436, Chile; anamariav.i@hotmail.com

**Keywords:** air pollution, source apportionment, cluster analysis, health risks, residential wood burning, sustainable urban development

## Abstract

Air pollution regulation requires knowing major sources on any given zone, setting specific controls, and assessing how health risks evolve in response to those controls. Receptor models (RM) can identify major sources: transport, industry, residential, etc. However, RM results are typically available for short term periods, and there is a paucity of RM results for developing countries. We propose to combine a cluster analysis (CA) of air pollution and meteorological measurements with a short-term RM analysis to estimate a long-term, hourly source apportionment of ambient PM_2.5_ and PM_10_. We have developed a proof of the concept for this proposed methodology in three case studies: a large metropolitan zone, a city with dominant residential wood burning (RWB) emissions, and a city in the middle of a desert region. We have found it feasible to identify the major sources in the CA results and obtain hourly time series of their contributions, effectively extending short-term RM results to the whole ambient monitoring period. This methodology adds value to existing ambient data. The hourly time series results would allow researchers to apportion health benefits associated with specific air pollution regulations, estimate source-specific trends, improve emission inventories, and conduct environmental justice studies, among several potential applications.

## 1. Introduction

Ambient air pollution is a major environmental risk worldwide. Current estimates of premature mortality brought by population exposure to ambient PM_2.5_ (particulate matter with aerodynamic diameter below 2.5 µm) vary between five million [1] to ten million [2] annual deaths. This burden of disease has prompted a worldwide effort to regulate ambient PM_2.5_ concentrations. A customary set of public policies targeting that overarching goal includes: ambient air pollution monitoring, emission inventories, air pollution modeling, health effects studies, and economic valuation of regulation measures (emission standards, ambient air quality standards, market-based instruments, etc.).

Ambient PM_2.5_ is directly emitted by traffic, industrial, commercial, and residential sources in urban zones, and by agriculture, mining, industrial, and natural sources elsewhere. However, the resulting ambient PM_2.5_ concentrations are far more complex to characterize. Some gaseous pollutants (sulfur and nitrogen oxides) undergo atmospheric oxidation, leading to sulfuric and nitric acids, which, in turn, react with ammonia to produce ammonium sulfate and nitrate particles, forming secondary PM_2.5_ [3]. Furthermore, volatile (VOC) and semi-volatile (SVOC) organic compounds are also oxidized in the atmosphere to become higher molecular weight compounds that condense to the particulate phase, generating secondary organic PM_2.5_ [4]. The large number of organic compounds released, along with their wide range of gas/particle phase partitioning, add further complexity to ambient PM_2.5_ [5]. Meteorology also plays a role in the previously mentioned chemical and physical processes, adding seasonality to ambient PM_2.5_ at different time scales: wind advection moves pollutant away from their sources, temperature enhances atmospheric chemical reactions and VOC emissions, solar radiation modulates photochemical processes, relative humidity modifies fugitive PM emissions, atmospheric stability controls the degree of vertical mixing of air pollutants, rainfall brings particles and gases down to the ground, etc.

Hence, a major hurdle in quantifying population exposure to ambient PM_2.5_ lies on the wide range of sources and processes that drive ambient PM_2.5_ concentrations, and their variability on different time scales: diurnal, weekly, seasonal, long-term. Although ambient PM_2.5_ monitoring is becoming widespread worldwide [6], a quantitative knowledge of the major sources accounting for those observed concentrations is still scarce. To obtain such knowledge, there are two customary modeling tools to conduct such a task: dispersion models and receptor models, which are described next.

Dispersion models (DM) use atmospheric emission inventories to model ambient PM_2.5_ at different spatial scales: urban, regional, continental, and global [7,8]. They use atmospheric modeling to simulate how those emission sources disperse, react, deposit on the surface as they are advected by the wind field, and mixed by atmospheric turbulence. In principle, these models provide a detailed spatial and temporal representation of ambient PM_2.5_ concentrations to evaluate regulatory policies at a national level. However, there are several sources of uncertainty in the emission inventories themselves and also in the parametrization of physical processes occurring in the lower atmosphere. Those two uncertainties propagate in the simulated ambient concentrations. For instance, urban zones in complex terrain pose a great challenge to current meteorological modeling tools [9], and the low-wind, stable meteorological conditions that drive air pollution episodes are difficult to be represented in current models [10]. In a review of photochemical and PM_2.5_ dispersion modeling applications in the US and Canada [11], the percentage of observed variance explained by the dispersion model (r^2^) varies between 0.2 and 0.6 for hourly ozone and between 0.1 and 0.6 for PM_2.5_ and its major species. These are interquartile ranges and, in some cases, the model performance is better. These results represent the typical uncertainty expected for this DM approach. 

Receptor models (RM) use chemical speciation of ambient PM_2.5_ samples, taken on filters (daily or longer time-accumulated samples), to estimate the major sources contributing to ambient PM_2.5_ mass concentrations [12,13,14,15,16]. They are based on the mass conservation principle applied to a set of tracer species (elements, isotopes, ions, carbonaceous aerosol, organic molecules). The required chemical speciation analyses are expensive, so most RM studies have been conducted in developed countries and have short-term results with a small proportion of long-term (multiyear) studies. A challenge for RM studies is to cover most urban zones in any given country to obtain a national level assessment of pollution sources to evaluate a regulatory policy. Regarding the uncertainty associated in RM results, one way of estimating it is to conduct a round robin exercise in which the very same data set is analyzed by different research groups. In one of these exercises conducted with a synthetic database [12], it was found that 87% of 344 RM source contribution estimates (SCE), generated by 47 participant groups, were within 50% of the correct result. Larger sources (SCE > 10% of total PM mass) were better quantified by RM, while smaller sources (SCE near 1% of total PM mass) were a challenge for most RM. In another European study (with an actual chemical speciation database), 38 RM were applied by 33 participant groups [15]. Additionally, 72% of the estimated time series of SCE were within 1-sigma uncertainty and 91% of the estimated SCE passed the z-score test. Therefore, receptor model results for SCE have a typical upper bound uncertainty of 50%.

A more recent generation of continuous ambient PM_2.5_ instruments has been developed. They provide continuous information on chemical speciation [17,18,19] and optical properties [20,21,22,23,24], which also leads to source apportionment, even at a finer temporal scale (minutes, hours) than filter-based RM applications, which typically correspond to daily or weekly samples. That kind of instrumentation is more expensive than traditional ambient monitors for regulated pollutants, so it is currently less widespread than RM applications.

The preceding paragraphs show that a quantitative apportionment of major sources of ambient PM_2.5_ in urban areas worldwide is yet to be accomplished, particularly in developing countries. This is the current knowledge gap.

With an ever-increasing global ambient air pollution monitoring by regulatory authorities, the possibility of extracting information out of these databases has been on the rise, estimating temporal trends in air pollution [25,26,27,28,29]. Furthermore, more information is expected to be collected due to the rise of low-cost ambient monitoring [30,31,32,33,34,35,36,37]. More insight into the likely sources of ambient PM_2.5_ and PM_10_ can be achieved by combining meteorological and air pollution measurements. Bivariate plots of ambient concentration as a function of wind speed and direction provide information on sources contributing to ambient concentrations [38,39]. For instance, tall stack and area sources contribute most when wind speeds are higher and lower, respectively [3]. One step further consists in performing a cluster analysis (CA) of this type of bivariate representation, so that major clusters contributing to ambient concentrations may be visualized and explored [40]. One limitation of this CA approach is that there is no simple recipe to know how many clusters should be chosen in a given analysis [41]. In other words, more information is needed to decide how many clusters may be resolved in any given set of ambient air pollution data.

In a review of CA applied to air pollution analysis between 1980 and 2019 [41], it is shown that most CA applications to surface observations (i.e., air quality monitoring) consisted in exploring spatial associations among monitor sites within the same city, region, or country. The temporal evolution of those clusters was focused on their seasonal variability. In one publication [42], chemical speciation of ambient PM_2.5_ was included in the cluster analysis to classify daily samples according to differences in PM_2.5_ chemical composition. Then back trajectory analyses were developed to estimate air mass origins associated with each resolved cluster. 

Our goal is to show that, by combining short-term RM analyses with results gotten from long-term (multiyear) CA, for the same location, it is possible to identify the major sources (clusters) in ambient air pollution data therein. With this approach, the RM results provide additional information to decide how many clusters should be considered in the CA, so both methods complement each other. We show a proof of the concept for this combined approach in three case studies of urban zones with widely different dominant sources and meteorological conditions. As far as we are aware of, this is the first time this joint analysis has been proposed for ambient air pollution data.

This combined analysis is facilitated with a set of simple rules to identify sources: PM_2.5_/PM_10_ ratios, dependence of ambient concentrations with temperature, wind speed and relative humidity, and the location of the monitoring site with respect to obvious sources: highways, industrial zones, etc. All these rules come from the literature of RM studies worldwide—see [16,43,44] for references —along with well-known results for the dispersion of stack emissions under different meteorological conditions [3] and the main features of motor vehicle emissions [45,46,47,48,49,50,51,52] as well as other combustion emissions [53,54,55,56,57,58].

The outcome of this proposed computational process is a long-term (multiyear), hourly time series of source contributions to ambient PM_2.5_ and PM_10_ (and other measured air pollutants as well) suitable for different research purposes: exploring association of health effects with specific sources (example: traffic), tracking source trends, evaluate effectiveness of sector regulations, constrain sector emissions through dispersion modeling, conduct environmental justice analysis, further analysis of long-range sources using backward wind trajectories, etc. A major implication of our work is that ambient air quality databases are a rich source for further air pollution analysis worldwide, especially for source apportionment of PM_2.5_ and PM_10_.

## 2. Materials and Methods 

### 2.1. Computational Methodology

We use the open access R software environment, including the openair package for air pollution analysis [59]. Openair has several dedicated functions for air pollution analysis, including the polarCluster function that groups ambient pollutant concentrations using bivariate plots. Briefly, polarCluster performs a k-means cluster analysis on the vectors [u, v, c] where u, v are the zonal and meridional wind components (sometimes referred to as the east and north wind components) and c as the pollutant concentration under analysis (PM_2.5_, PM_10_, etc.). Since all these variables have different scales, they are all standardized before the k-means algorithm proceeds [40]. This k-means clustering algorithm is well-known and has been extensively used in air pollution analyses [41], as mentioned above.

Regarding the receptor model methodology, we use already published RM results for two Chilean cities: Santiago and Temuco [60,61]. For Santiago, USEPA Positive Matrix Factorization (PMF, version 5.0) was the RM used [62], while for Temuco, the USEPA Chemical Mass Balance (Version 8.2) was the RM used [63]. Both models solve the following mass balance equation for p sources [13].
(1)$$X_{ij}=\overset{p}{\underset{k=1}{\sum }}g_{ik}f_{kj}+e_{ij}$$
where *X_ij_*, *g_ik_*, and *f_kj_* are matrices whose entries are the j-th species mass concentration measured in the i-th sample, the mass concentration from the k-th source contributing to the i-th sample, and the j-th species mass fraction in the k-th source, respectively. *p* is the total number of resolved sources. The residuals *e_ij_* are assumed random and normally distributed.

The Chemical Mass Balance (CMB8.2) model solves Equation (1) for the case when the source profiles {*f_kj_*} have been experimentally measured for the k-th source. Hence, Equation (1) is solved for {*g_ik_*} using an effective variance least squares approach [64]. The Chemical Mass Balance model assumes that all major sources in the study area have been identified and included in its input. This model is usually run for different combinations of sources until a satisfactory solution is achieved in terms of the percentage of observed variance explained by the model.

The Positive Matrix Factorization (PMF5) model solves Equation (1) without making any assumption regarding source profiles {*f_kj_*}. However, in such a case, there are more unknowns than equations in Equation (1). This means additional information must be supplied into the model [65]. Usually, source compositions {*f_kj_*} and contributions {*g_ik_*} are required to be non-negative. This is the case of software PMF5, which minimizes the following function.
(2)$$Q=\overset{n}{\underset{i=1}{\sum }}\overset{m}{\underset{j=1}{\sum }}{\left[\left(X_{ij}-\overset{p}{\underset{k=1}{\sum }}g_{ik}f_{kj}\right)/\sigma_{ij}\right]}^{2}$$
where *σ_ij_* is the estimated uncertainty in the j-th species at i-th PM sample. In this approach, all observations are individually weighted by their respective uncertainties. The above minimization is carried out including the previously mentioned non-negative constrains upon compositions {*f_kj_*} and contributions {*g_ik_*}.

More details on the information used for each RM application (sampling period, chemical analyses, etc.) are provided in the respective references above [60,61]. 

### 2.2. Case Studies

We have selected, as case studies to test the proposed methodology, three urban zones in Chile with widely different meteorological conditions and dominant sources in which RM results are known. Calama (22°27’S, 68°55’W, population 2017: 158,600 [66], elevation: 2500 m) is a city located within the Atacama Desert (Koppen-Geiger classification BWk [67]) and the city is close to a large mining district. The major pollution problem is ambient PM_10_, given the desert conditions and rather strong winds that promote road dust resuspension by traffic and aeolian dust blown from the desert environment. Santiago (33°28’S, 70°40’W, population 2017: 6.85 million [66], elevation: 600 m) is the capital city, and extends along a valley surrounded everywhere by ranges and the Andes cordillera to the east. The climate is a warm temperate with dry summer (Koppen-Geiger classification CSb). Air quality regulations in Santiago were the first to be enacted in Chile and they have been successful in curbing down ambient PM_2.5_ [68,69,70,71]. Nonetheless, ambient PM_2.5_ in Santiago currently exceeds World Health Organization (WHO) guidelines and Chilean ambient air quality standards (AAQS), and major PM_2.5_ sources are traffic, industry, commercial, and residential sources. The third case study is the city of Temuco (38°44’S, 72°35’W, population 2017: 308,600 [66], elevation: 350 m) with a warm temperate fully humid climate (Koppen-Geiger classification Cfb). In this city, residential wood burning emissions rise in colder months, leading to severe ambient PM_2.5_ concentrations [60] that increase indoor concentrations as well [72]. 

Ambient air pollution (PM_10_, PM_2.5_, CO, SO_2_, NOx) and surface meteorological measurements were downloaded from Chile’s Air Quality Information System (https://sinca.mma.gob.cl/). Table 1 shows the summary of information used to conduct the analyses, where one monitoring station per city was chosen. Figure 1 shows the locations of the three monitoring sites used in our analyses. Data were screened for obvious outliers and these were removed before the computational analyses. 

### 2.3. Simple Comparison Rules

To map the outcome of the CA results with the RM results for the same site, we propose to use a set of rules. Some of these rules have been traditionally used in identifying RM results, such as looking at the weekly seasonality of source contributions to identify traffic or industrial sources [61,70,71,73]. The former is lower over weekends while the latter is not. Other rules use the hourly resolution of CA results to generate scatter plots and time-variability plots of ambient PM_2.5_, PM_10_, and gases (CO, NOx, SO_2_), stratified by cluster, to check for specific patterns. For instance, large industrial stacks contribute more to ambient concentrations when wind speeds are higher [40] and ship emissions contribute more when ambient temperature rises [74]. A key result from scatter (X-Y) plots of air pollutants is the graphical interpretation of source compositions as ‘limiting edge lines’ [13,75], which define an upper (lower) edge line with the highest (lowest) Y/X ratio. Furthermore, the ratio PM_2.5_/PM_10_ is different for combustion particles than for mechanically-generated particles [3]. The proposed rules and their rationale are detailed next. 

Residential wood burning (RWB) sources exhibit a high PM_2.5_/PM_10_ ratio, typically 0.7 or higher, with features of a single source in a PM_2.5_–PM_10_ scatter plot, that is, most points lie along a straight line. RWB contributions show the highest seasonality off all resolved clusters (sources) peaking on colder months. This is a consequence of RWB emissions being driven by increasing space heating demand, so they increase in colder months whereas traffic and industrial sources remain constant all year long. Likewise, hourly RWB contributions increase when ambient temperature decreases. With respect to relative humidity (RH), RWB contributions tend to increase at higher RH, while other area sources (like fugitive dust) decrease as RH increases. RWB contributions tend to peak near midnight in colder months, unlike traffic sources that peak earlier in the evening. On a weekly basis, RWB contributions decrease less over weekends than traffic contributions do.Traffic sources display a PM_2.5_–PM_10_ scatter plot with a cloud of points bounded by ‘limiting edge lines’: an upper edge line close to a 1:1 line, characteristic of exhaust emissions, and a lower edge with a small PM_2.5_/PM_10_ ratio, typical of non-exhaust traffic emissions (i.e., road dust [76]). This is a typical signature whenever a pair of sources contribute to ambient concentrations [13,75]. Traffic contributions universally decrease on weekends, unlike industrial (or mining) sources, which are steady all year long. On a diurnal basis, traffic sources show distinctive morning and evening rush hour peaks. When plotted against wind speed, traffic sources display a negative correlation, explained by the better ventilation conditions brought by higher wind speeds. In contrast, industrial sources do not show a clear correlation or sometimes display a positive correlation because higher wind speeds brought high stack emissions down to the ground in unstable atmospheric conditions.Industrial sources appear as hot spots in the CA outcome, associated with specific wind directions and high wind speeds, when tall stack contributions are relevant through fumigation processes [40]. An inspection of CA results for sulfur dioxide (SO_2_) helps to clarify the location and contributions of those industrial spots to ambient PM_2.5_. Since, under stable atmospheric conditions (and lower temperature and wind speeds), stack plumes will rise, their contributions to ambient SO_2_ will be negligible and, therefore, traffic contributions will dominate under these circumstances. Conversely, under unstable atmospheric conditions, higher wind speed and temperatures will promote contributions from stacks, which will be the dominant ones for SO_2_. In coastal areas, this mechanism will work the same way for SO_2_ from ship emissions [74].Aeolian dust sources appear only at high wind speeds, and, to resolve them in the CA, the number of clusters needs to be increased until those sources emerge, provided we know they are at play in the study zone. As the number of clusters increases in the CA, more intermittent sources are likely to show up. Most of these intermittent sources contribute little to no ambient PM_2.5_ on a long-term basis. However, they are unlikely to show up in the RM results because RM have difficulties in resolving intermittent sources with low contributions to ambient PM_2.5_ [12]. Nonetheless, they might indicate long-range, regional sources arriving to the monitoring site. Thus, these may be analyzed on their own using backward wind trajectories [77,78] to confirm their identity (natural or anthropogenic). Ubiquitous area sources (like traffic) would be split by wind direction sectors as the number of clusters increases. They will all have the features presented in rule 2 above. Additional gaseous measurements will provide further insights into sources’ identities, comparing how those concentrations distribute across clusters. In the case of nitrogen oxides (NO, NO_2_, NOx = NO + NO_2_), a cluster consisting of cleaner air masses (coming from the ocean, for instance) will have low NO_2_ and NOx values, while aged, long-range anthropogenic regional sources in another cluster will display larger NO_2_/NOx ratios. In the case of carbon monoxide (CO), this pollutant should be apportioned mostly to traffic sources, and, hence, it would belong to traffic-related clusters. The exceptions are zones when RWB or some industrial sources are relevant. These will apportion CO as well, particularly in colder months (RWB) or under high wind speeds (industrial stack sources). SO_2_ is a good tracer for large industrial sources such as copper smelters or coal-fired thermal power generation units. They may also be used as a tracer of ship emissions.

## 3. Results

### 3.1. Results for Calama 

In this city in an arid environment, annual ambient PM_2.5_ concentrations are below 10 µg/m^3^, that is, ambient PM_2.5_ satisfies current WHO guidelines [79]. Thus, we focus on ambient PM_10_ as the pollutant of concern, considering that fugitive sources, such as those windblown from desert surroundings and road dust, should be relevant. Both sources are also hard to estimate, given the intermittence of physical processes that lead to high wind speeds (gustiness conditions) in the former source, and the heterogenous processes responsible for surface dust loading on the city’s streets in the latter source [51].

For this city, no RM results have been published. Nonetheless, available urban emission inventories for Calama [80] estimate that road dust (or traffic non-exhaust emission) is the dominant PM_10_ emission source, being 88% of urban emissions (Appendix A). Therefore, traffic emissions dominate PM_10_ emissions. However, no estimation of windblown dust is available for this study zone. We conduct the analysis for the monitoring station ‘Centro’ in Calama (see Table 1 for further details).

Figure 2 shows the outcome of the polarCluster routine when applied to ambient PM_10_ data gathered from October 2012 through August 2020. The clusters associated with windblown dust emerge with high wind speeds and wind directions between 270° and 360°, and stay the same for solutions of CA with 5–9 clusters. The other clusters include low wind speed conditions and they split into several wind direction sectors as the number of clusters increases. For simplicity, we choose a solution with five clusters and explore the results using the simple rules presented in Section 2.3 to analyze the results.

Figure 3 shows the source contributions associated with the five-cluster solution for PM_10_. Clusters 1 and 2 are the dominant ones, followed by clusters 4, 3, and 5. The first analysis is performed looking at the time variability of this five-cluster solution. Appendix A shows that clusters 3 and 5 present higher contributions between noon and 6 PM when wind speeds are higher in the region. Appendix A shows scatter plots of ambient PM_10_ versus wind speed. PM_10_ concentrations from clusters 3 and 5 clearly increase with wind speed, which is distinctive of Aeolian emissions. By contrast, other clusters’ contributions have little increase or decrease with wind speed (see R^2^ values for instance). Cluster 3 comes from W and WNW directions and cluster 5 from NNW winds. Cluster 3 is characteristic of upslope, anabatic winds that develop all year long given the strong solar irradiation and extreme soil dryness [81]. Cluster 5 is characteristic of mountain synoptic conditions that happen in the austral fall and winter seasons, when the Pacific subtropical high weakens, favoring a higher frequency of north winds. Nonetheless, both windborne dust sources are minor contributors to a long-term average PM_10_, as can be seen in Figure 3.

Figure 4 shows the PM_10_ time variability for clusters 1, 2, and 4. It is clear from this figure that these three clusters correspond to traffic sources: their ambient contributions have distinctive peaks during morning and evening traffic rush hours, and they significantly decrease over the weekends. The same behavior can be seen in Figure 5, where the NOx time variability is plotted for clusters 1, 2, and 4. Figure 6 shows scatter plots of PM_2.5_ and PM_10_ by cluster. In clusters 1, 2, and 4, the points lie between two limiting ‘edge lines’: one upper edge with high PM_2.5_/PM_10_ ratio (traffic exhaust) and a lower edge with a low PM_2.5_/PM_10_ ratio (lower that 0.1), characteristic of non-exhaust traffic emissions [46,51]. Thus, any point in the plot for those three clusters can be regarded as an air mass that arrives to the monitor site with a mixture of those two traffic emissions, as it is customarily presented in the receptor modeling literature [13,75].

### 3.2. Results for Temuco 

In this city located in a wet temperate climate, RWB is the major source of ambient PM_2.5_. RM results show that, in the winter season, RWB contributions to ambient PM_2.5_ vary between 70% and 100% [60]. These RM results were found for a monitoring site located 700 m NW of the air quality monitoring site that we analyze here (denoted as LE by its Spanish name ‘Las Encinas’). The data cover the period from January 2009 through August 2020. In this case study, we use temperature instead of wind speed for conducting the CA. Had we chosen the traditional CA, we would have found difficulties in interpreting the resulting clusters because of an overlapping of high PM_2.5_ contributions at low wind speeds, which mixes traffic and RWB contributions (results not shown). By using temperature as an input variable, we are able to extract the RWB contribution and separate it from the traffic contribution. This approach works well precisely because the highest RWB impacts occur at nearly midnight in colder months when traffic sources are minimal and RWB emissions are the highest. Furthermore, since under those stable atmospheric conditions, wind direction is highly variable because of wind meandering effects, RWB contributions should come from all wind directions, defining a single cluster enclosing the origin of coordinates in the bivariate plot. This geometric feature helps in identifying RWB sources, even when they are not the dominant ones (see Section 3.3). 

Figure 7 shows the cluster analysis results for two to eight clusters in Temuco, Las Encinas (LE) site. It can be clearly seen that a ‘central cluster’ develops for solutions with three or more clusters. For seven or more clusters, this central cluster splits itself into a pair of low-temperature and high-temperature clusters. They both correspond to RWB (results not shown here). For simplicity, we choose a three-cluster solution to identify the sources.

Figure 8 shows the monthly source contributions of the three-cluster solution found in Temuco. Source 3 is the dominant one, followed by sources 1 and 2. Figure 9 shows the time variability of those three clusters. Clusters 1 and 2 show lower seasonality than cluster 3, which has dominant contributions near midnight. Appendix A shows the pollution rose by the cluster. Cluster 3 has contributions for low wind speeds and from all wind directions, while clusters 1 and 2 are each associated with a set of narrow wind directions. Figure 10 displays PM_2.5_-PM_10_ scatter plots. Clusters 1 and 2 have two clear limiting edge lines—characteristic of traffic sources—while cluster 3 points lie along a straight line with a high slope near 1. Appendix A shows CO time variability by cluster. The cluster patterns mimic the ones already shown in Figure 9 for PM_2.5_. Figure 11 show scatter plots of PM_2.5_ against the temperature. Cluster 3 has a different behavior and higher PM_2.5_ concentrations than in clusters 1 and 2. Therefore, the CA results indicate that residential wood burning is the dominant source of ambient PM_2.5_, which is followed by traffic sources (clusters 1 and 2) with lower contributions.

We now turn to a quantitative comparison of the above CA results with RM results obtained for the period July–September 2014 at a monitoring site 700 m NW of the LE site. Table 2 shows a comparison of the average sources identified with CA and those resolved using the chemical mass balance RM (CMB8.2) with organic molecular markers [60]. The molecular markers measured in ambient PM_2.5_ samples were used to apportion organic carbon (OC) among different combustion sources (gasoline, diesel, wood, coal, natural gas) using Equation (1) with known source profiles {*f_kj_*}. Afterward, organic source contributions to PM_2.5_ mass were calculated from those source contributions to OC and specific OC/PM_2.5_ mass ratios for each source [82,83,84,85]. 

For the CA traffic sources, we compare them with the sum of diesel exhaust emissions, vegetative detritus (both resolved by CMB8.2), and dust (sum of Al, Si, Fe, Ca, and Ti oxides) to consider the non-exhaust contributions included in the CA (i.e., road dust). For the residential wood burning source, the comparison is more elaborated. We have found that, in Temuco, coal combustion is also used for space heating (identified using picene as organic tracer), so we have added RWB contributions (identified using levoglucosan as an organic tracer) to those from residential coal combustion since both processes happen under the very same environmental conditions. Furthermore, there is a substantial contribution of secondary organic aerosol (SOA) in Temuco, which is ascribed to the inefficient burning of wood. This combustion process is known to release semi-volatile organic compounds [53], which quickly oxidize and, thus, generate secondary organic aerosols [86]. The CMB8.2 RM cannot resolve secondary sources, so the unresolved organic carbon (OC) is denoted as ‘Other OC’. This ‘Other OC’ is identified as SOA by its high correlation with water soluble organic carbon (WSOC), indicating a high degree of molecular oxygenation. Therefore, we added the corresponding SOA contribution to the RWB (and coal combustion contribution) to get the ‘RWB RM’ entry in Table 2. 

We show in Table 2 the previously mentioned comparisons for the 8-week ambient monitoring campaign reported in Reference [60]. For every weekly sample, we computed the average CA contributions for both sources (Traffic: clusters 1 and 2, RWB: cluster 3), considering the same five sampling days per week as in the ambient measurement campaign. Standard errors for all estimates (from CA and RM results) were computed from error propagation. For both major sources that can be resolved with CA, the agreement is good with very similar winter average results. The zero value for one week in the traffic CA contribution is a result of trying to identify a weak signal in a data set dominated by a single source (RWB in this case). This is the usual outcome both in CA and RM analyses.

Therefore, the comparison results in Table 2 show that the proposed CA analysis is able to capture the major sources contributing to ambient PM_2.5_ in this case study of a zone dominated by residential wood burning emissions. For the (smaller) traffic contributions, the methodology has difficulties in capturing the temporal variation, even though, on average, the results are similar to the RM results.

### 3.3. Results for Santiago 

In this case study of a large metropolitan area, we focus the analysis on a monitoring site located near the east edge of the city at a higher elevation in Santiago’s basin (henceforth, denoted as LAC, which is short for its Spanish name ‘Las Condes’). For that site, a previous RM study [61] has shown that traffic sources and residential wood burning are the dominant ones, although regional sources also contribute to ambient PM_2.5_. That study was conducted with another RM, Positive Matrix Factorization (PMF), using elemental concentrations in ambient PM_2.5_ daily samples as input data.

Figure 12 shows the results of CA for 2 to 10 candidate clusters, using temperature instead of wind speed as input variable, to resolve RWB contributions. A central cluster with the lowest ambient temperatures shows up for solutions with seven or more clusters, suggesting this is the RWB source. We choose an eight-cluster solution and we present these results here. Figure 13 shows the source contributions brought by this eight-cluster solution. Clusters 3 and 4 are the ones with the largest source contributions, followed by cluster 7 (that groups the lowest ambient temperatures) and clusters 1 and 6. The rest of the clusters are of minor relevance. Appendix A shows that cluster 3 has predominant WSW directions while cluster 4 includes ENE and E directions. Figure 14 shows the PM_2.5_ time variability but only for the five major contributing clusters. It can be seen that cluster 4 has morning and evening peaks, coincident with rush hour traffic conditions. In cluster 3, its contribution increases from morning to early afternoon, indicating the arrival of traffic contributions from the city, brought by anabatic winds. This rise is followed by a decline of contributions later in the evening. This daylight behavior of anabatic winds and pollution transport toward the east side of the city has been recently measured and modeled for black carbon particles in Santiago [87]. Cluster 7 is the only one that does not decrease over weekends, unlike the other clusters shown in Figure 14, which decrease over weekends. This temporal pattern supports the identification of cluster 7 as the RWB sources. 

Figure 15 shows PM_2.5_-PM_10_ scatter plots by cluster, showing that, in cluster 7, the data have the highest slope of all, suggesting that this is the RWB source. For clusters 1–5 and 8, the scatter plots show that the respective data points have upper and lower limiting edge lines, as expected for traffic sources.

To better identify this eight-cluster solution, we present in Table 3 the monthly average PM_2.5_ contribution by cluster, for year 2004 for which we have RM results in this same monitoring site. 

From Table 3, we can see that clusters 4, 7, and 8 increase their contributions during the fall and winter season, whereas clusters 1, 2, and 3 show minimum contributions in those colder seasons. The reason for this different behavior has to do with meteorological conditions. In the fall and winter seasons, subsidence conditions promote a low level of thermal inversion layers over Santiago’s valley, leading to shallow planetary boundary layers (PBL) [88] and blocking transport of emissions from Santiago’s lower valley (dominant wind direction for clusters 1, 2, and 3 as seen on Appendix A). This also explains why local sources (clusters 4, 7, and 8) increase their contributions during the fall and winter. This topography-induced effect has been reported before for total ambient PM concentrations [89] and, more recently, in the simulation of black carbon transport from Santiago toward the Andes mountains east [87].

Regarding cluster 6, it has a different time variability as compared with the rest of the resolved clusters with contributions peaking in the afternoon and increasing in the fall and winter seasons. Therefore, they do not come from Santiago’s lower valley. RM results [61] indicate that regional anthropogenic sources contribute to ambient PM_2.5_ at the monitoring site, diagnosed by the presence of arsenic and sulfur in ambient PM_2.5_ samples. To check whether cluster 6 could represent those regional sources, Appendix A shows the source contributions to ambient SO_2_. It can be seen that cluster 6 SO_2_ contributions show up all year long, so they cannot come from Santiago’s lower valley. Besides, Appendix A shows that cluster 6 has W and WSW wind directions, suggesting that those regional sources are located west of Santiago. This result agrees with the location of regional sources of arsenic and sulfates identified in Reference [61] using backward trajectory analyses. Therefore, we conclude that cluster 6 can be identified as representative of regional sources of PM_2.5_.

In order to make comparisons between the CA results and the RM results, we need to consider the limitations of both analyses. The RM results were obtained using Positive Matrix Factorization software and elements as tracers. Organic tracers were not included and this is a limitation. For instance, more recent results [90,91] have shown that secondary organic aerosols are relevant in the spring and summer seasons in Santiago. This secondary PM_2.5_ was not resolved by the PMF solution including only elements. Another limitation of the PMF solution is the use of potassium as a tracer of wood burning. Potassium may be a reasonable tracer in fall and winter seasons, but not so specific in the spring and summer when soil dust becomes relevant in Santiago’s semi-arid climate [70]. Because of these issues, we have decided to compare CA and RM results only for the months from May through August 2004. Table 4 shows such a comparison. We have grouped clusters 3 and 4 as traffic sources, cluster 7 as the RWB source, and cluster 6 as regional sources. The smaller contributions from other clusters have not been considered, given the limitations of both CA and RM to resolve smaller (or intermittent) sources. Since the filter-based RM results consider only a subset of days per month, these specific days have been extracted from the CA results to compute comparable averages. 

From the results in Table 4, it can be seen that CA tends to overestimate the RM result for traffic contributions (which includes traffic and soil dust contributions) but CA results for RWB and regional sources contributions that are below those estimated by the RM in the very same monitoring site. In fact, the sum of CA estimates in Table 3 has an average of 33.3 (µg/m^3^), while the corresponding average of RM results is 41.3 (µg/m^3^). The reason for this discrepancy is ascribed to the different PM_2.5_ measurement techniques: filter samples were taken in low-volume dichotomous samplers (Andersen Instrument, Inc, Smyrna, GA, USA, 15 L/min) while continuous measurements were made with a Tapered Element Oscillating Microbalance equipment (TEOM, Rupprecht & Patashnick, MA, USA). The latter instrument is known to present negative artifacts (i.e., underestimation of PM_2.5_ concentrations) due to partial volatilization of sampled PM_2.5_ in the TEOM’s heating inlet used to dry the samples [92]. We think this instrument artifact explains why filter-based RM results are higher than the CA results reported here. This effect explains the lower contributions found for RWB in the CA because RWB sources have a larger fraction of volatile compounds emitted, as compared with other combustion sources [53]. We do not have such an issue in the case of Temuco because, in that monitoring site, a beta-attenuation monitor (BAM, MET-ONE 1020, Met One Instruments Inc., Grants Pass, OR, USA) has been used to measure PM_2.5_ and PM_10_.

## 4. Conclusions

We have proposed that a CA approach, applied to ambient air pollution and meteorological data, provides a source apportionment of ambient PM_2.5_ and PM_10_ on an hourly basis. In order to achieve this result, we need to compare the outcome of the CA with RM results for the same monitoring site—or using an emission inventory in case no RM result is available—to identify the major sources (clusters) at play on a given zone. We have shown that our rule-based CA works for three different case studies in Chile: a city in a warm, desert region (Calama), another one dominated by RWB pollution in a cold, wet region (Temuco), and a large metropolitan area in a semi-arid region (Santiago).

In the case of Calama, the CA for ambient PM_10_ is able to resolve local sources (traffic) and windblown dust coming from the nearby desert environment. Both are fugitive sources, which are difficult to estimate, because of the amount of information required on each case, such as particle size distributions. CA results indicate that traffic sources dominate ambient PM_10_ concentrations, as suggested by the emission inventory of PM_10_ sources for that city. CA results show that, over the long-term, outbursts of windblown dust are not significant within the city. This is relevant for policy purposes. The evolution of traffic contributions in Calama (Figure 3) suggests that the ongoing street sweeping program has been a successful one with a clear decreasing trend. This is an example of how a sector-specific regulation in the city may be evaluated. Furthermore, within the traffic source, ambient PM_10_ data can be regarded as a combination of exhaust and non-exhaust emissions, providing additional insights such as whether new exhaust emission standards for motor vehicles have curbed down exhaust emissions.

In the case of Temuco, with dominant RWB contributions to ambient PM_2.5_, the use of ambient temperature instead of wind speed improves the apportionment of all major sources of ambient PM_2.5_. RWB contributions show up in the bivariate polar plots as a ‘central’ cluster because, under stable atmospheric conditions (with lowest ambient temperatures and wind speeds), wind direction is highly variable. Hence, the monitor site will sample air masses from all wind directions. CA source contributions for RWB sources are statistically comparable with RM results obtained for RWB sources in a short-term campaign in 2014.

For the large metropolitan area of Santiago, the CA methodology resolved RWB, traffic, and regional sources of ambient PM_2.5_. The CA analysis again required using ambient temperature to resolve the RWB contribution. The seasonality of the resolved clusters showed a distinctive effect brought by topography: Santiago’s plume does not reach the eastern side of the city in fall and winter seasons due to low PBL depths in colder months. This geographic condition provided an additional criterium to identify local and non-local traffic sources. However, the RM resolved source contributions (short-term campaign in 2004) were higher than the CA resolved counterparts. This is a result of an underestimation of ambient PM_2.5_ monitoring brought by volatilization losses in the continuous TEOM PM_2.5_ monitor. 

The rule-based CA presented here generates added value to existing ambient air pollution databases. Regarding RM results to complement the CA analysis, methods that use specific source tracers (like organic molecular markers) are preferable. 

The rule-based CA results may be applied to the following analyses:Identifying specific meteorological conditions leading to high PM_2.5_ concentrations, like windblown dust in arid regions.Provide long-term time series of source contributions to constrain emissions through DM applications to improve emission inventories.Tracking source trends and assess efficiency of specific regulations.Conduct environmental justice studies with the aid of low-cost air pollution monitoring (citizen science).Identify intermittent sources contributions, which may be further pinpointed using backward trajectory analysis. Conduct epidemiological studies to find associations between health effects and exposure to a single PM_2.5_ source such as traffic.Help in analyzing massive databases coming from state-of-the-science continuous monitors such as time-of-flight mass spectrometers measuring aerosols or VOC, multi-wavelength aethalometers, etc. 

The proposed rule-based CA analysis has limitations, though. The methodology works well for resolving the larger sources at play in a city, but smaller sources remain a challenge. 

## Figures and Tables

**Figure 1 ijerph-17-08455-f001:**
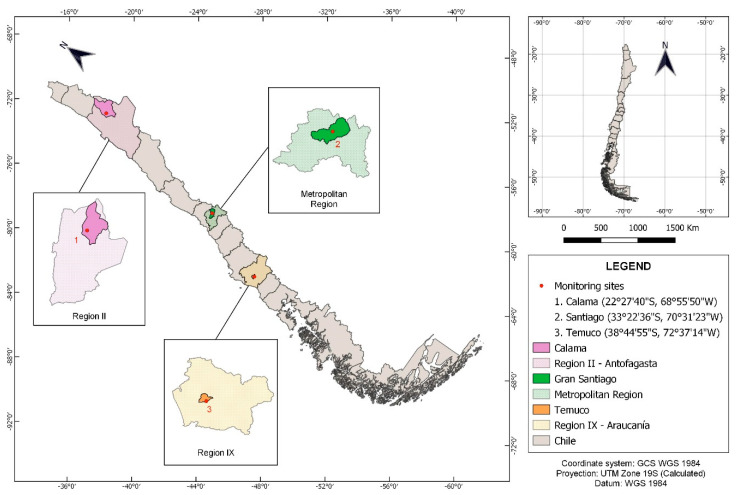
Geographical locations of the three cities analyzed.

**Figure 2 ijerph-17-08455-f002:**
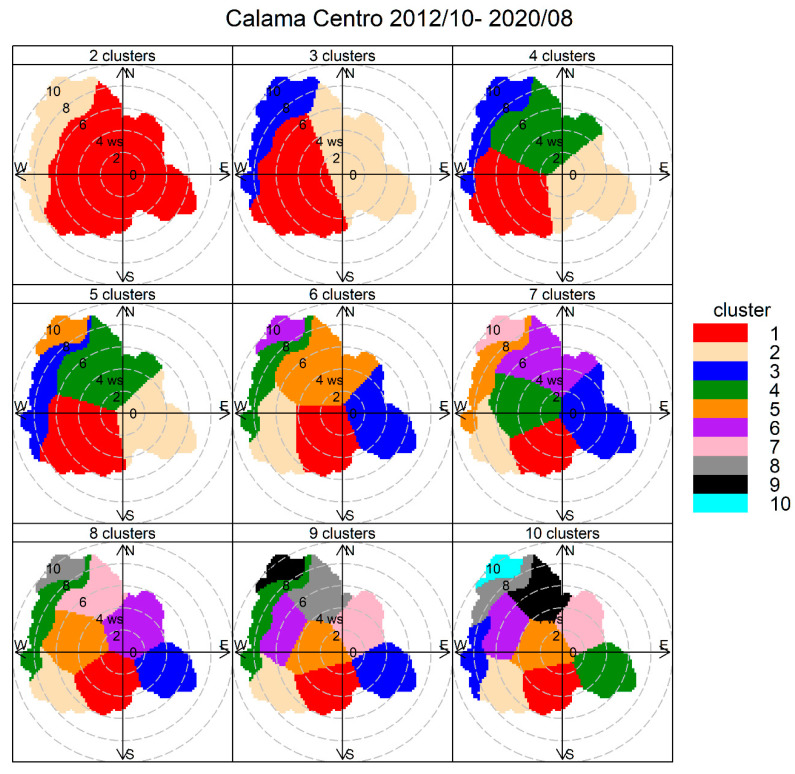
Cluster analysis source apportionment results for ambient PM_10_ at Calama, 2012/10–2020/08, for 2–10 clusters solutions. The numbers stand for wind speed levels (m/s).

**Figure 3 ijerph-17-08455-f003:**
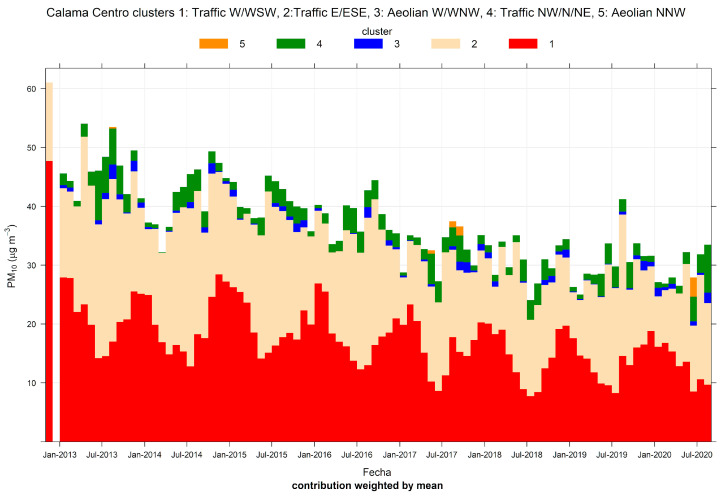
Cluster analysis source apportionment results for ambient PM_10_ at Calama, 2012/10–2020/08, for a 5-cluster solution.

**Figure 4 ijerph-17-08455-f004:**
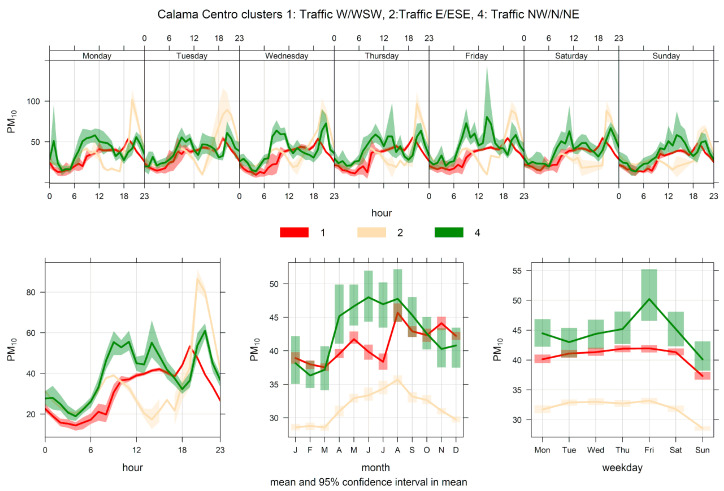
Time variation plot for cluster analysis source apportionment results for ambient PM_10_ at Calama, 2012/10–2020/08, for a five-cluster solution. Only clusters 1, 2, and 4 are plotted for clarity’s sake.

**Figure 5 ijerph-17-08455-f005:**
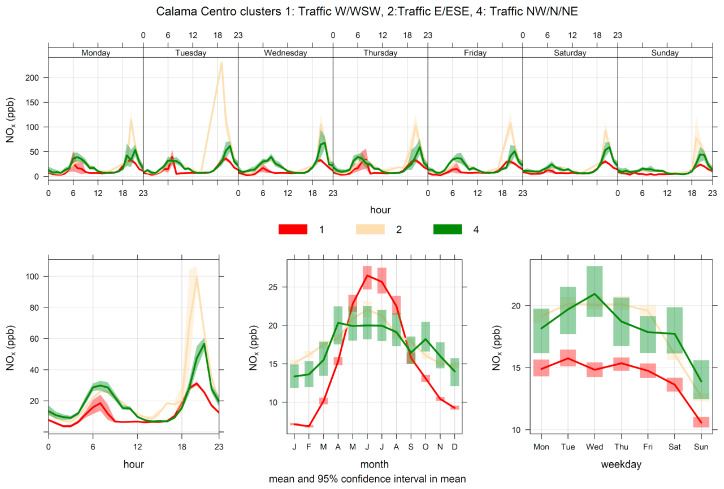
Time variation plot for cluster analysis source apportionment results for ambient NOx at Calama, 2012/10–2020/08, for a five-cluster solution. Only clusters 1, 2, and 4 are plotted for clarity’s sake.

**Figure 6 ijerph-17-08455-f006:**
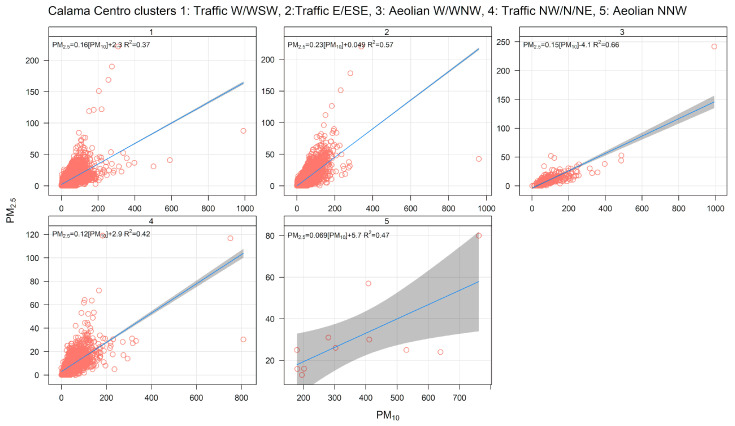
PM_2.5_–PM_10_ scatter plot by cluster for a five-cluster solution for Calama.

**Figure 7 ijerph-17-08455-f007:**
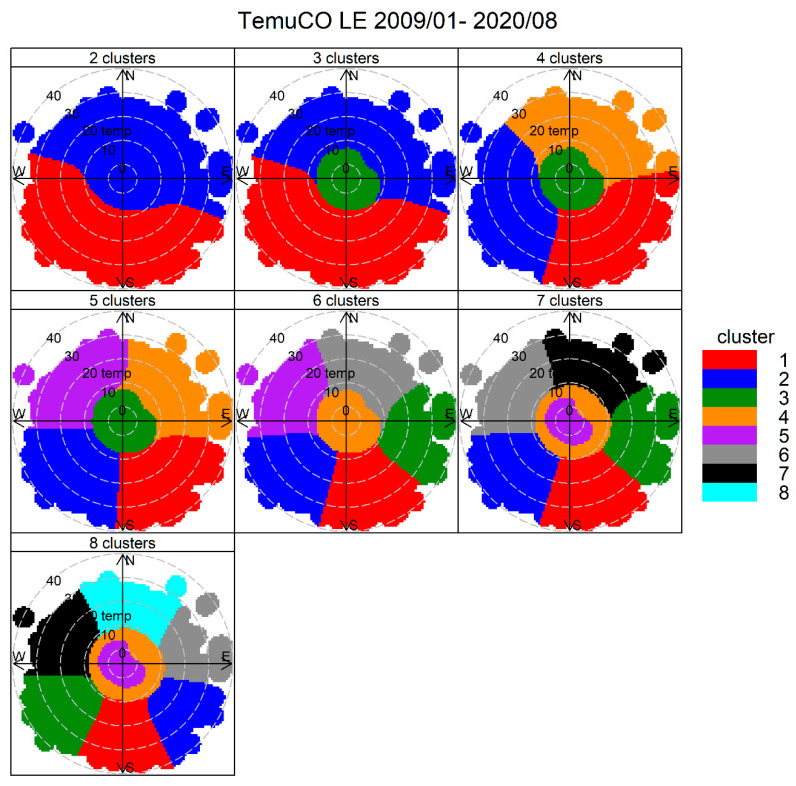
Cluster analysis source apportionment results for ambient PM_2.5_ at Temuco, 2009/01–2020/08, for 2–8 cluster solution.

**Figure 8 ijerph-17-08455-f008:**
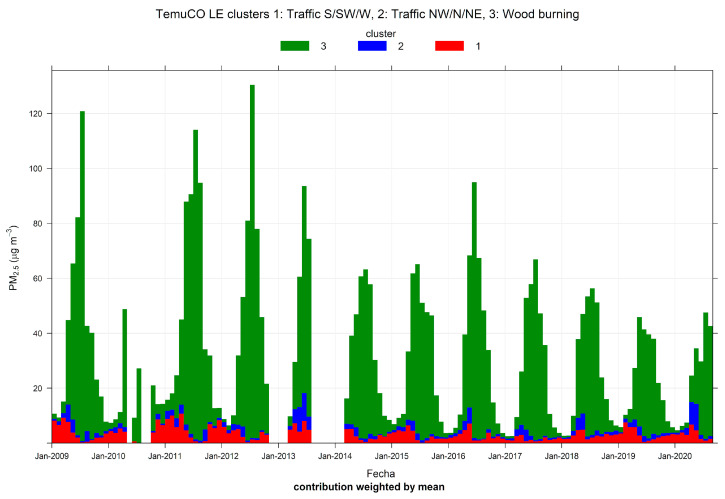
Cluster analysis source apportionment results for ambient PM_2.5_ at Temuco, 2009/01–2020/08, for a three-cluster solution.

**Figure 9 ijerph-17-08455-f009:**
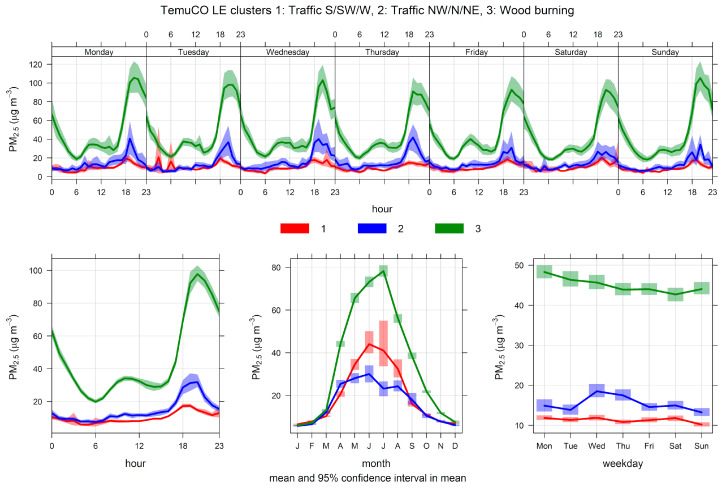
Time variation plot for cluster analysis source apportionment results for ambient PM_2.5_ at Temuco, 2009/01–2020/08, for a three-cluster solution.

**Figure 10 ijerph-17-08455-f010:**
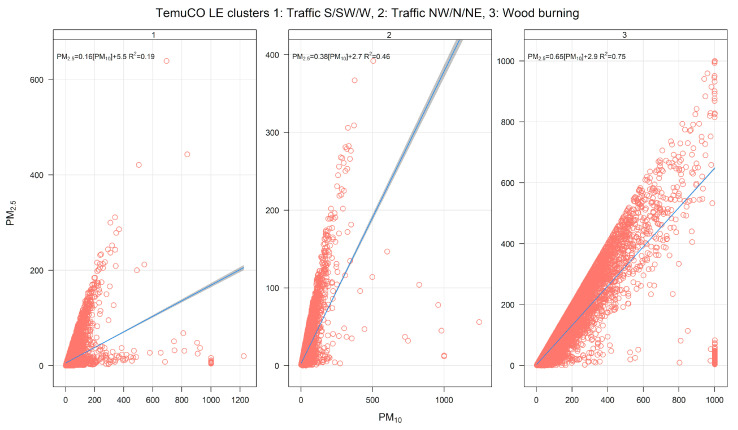
PM_2.5_–PM_10_ scatter plot by cluster, for a three-cluster solution for Temuco.

**Figure 11 ijerph-17-08455-f011:**
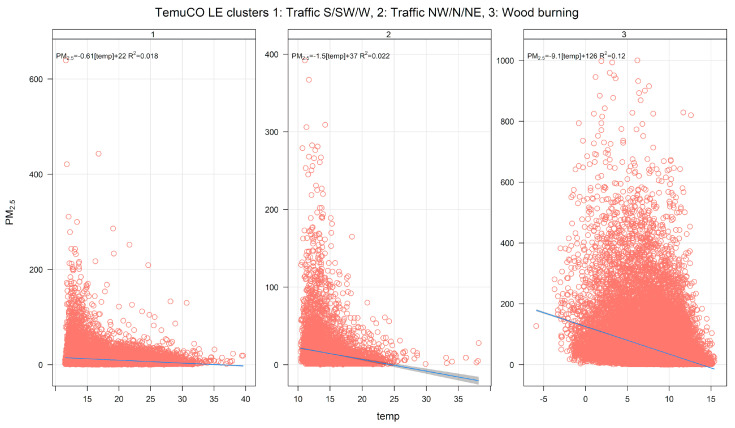
PM_2.5_-temperature scatter plot by cluster for a three-cluster solution for Temuco.

**Figure 12 ijerph-17-08455-f012:**
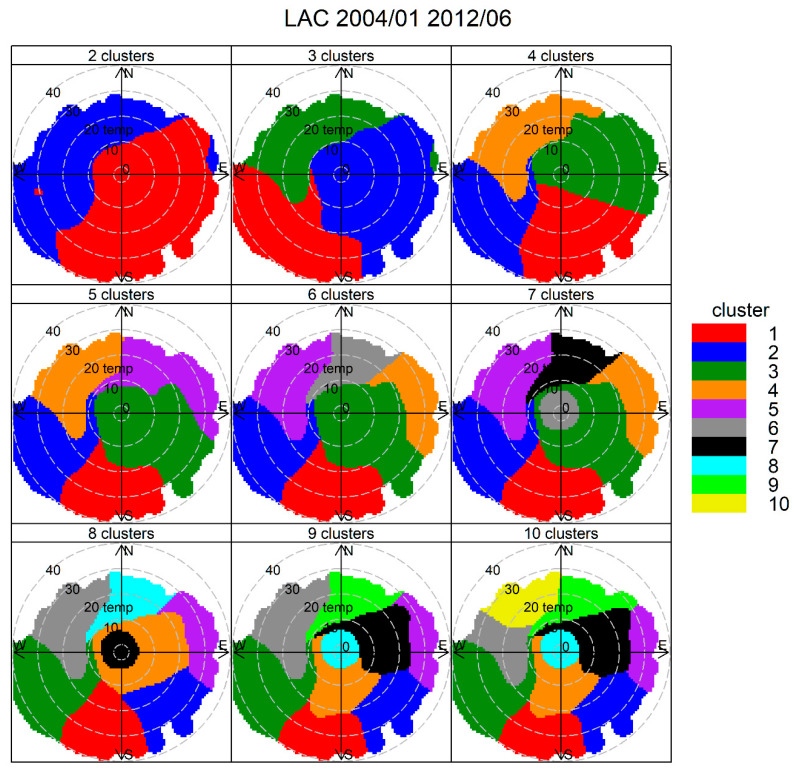
Cluster analysis source apportionment results for ambient PM_2.5_ at Santiago, 2004/01–2012/06, for 2–10 cluster solutions.

**Figure 13 ijerph-17-08455-f013:**
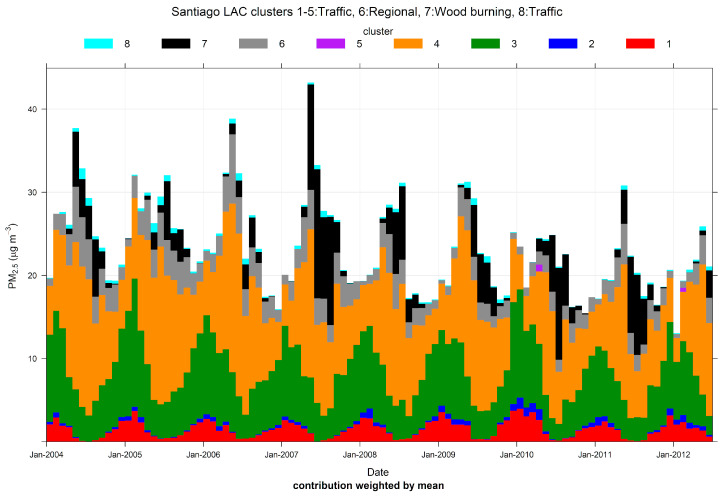
Cluster analysis source apportionment results for ambient PM_2.5_ at Santiago, 2004/01–2012/06, for an eight-cluster solution.

**Figure 14 ijerph-17-08455-f014:**
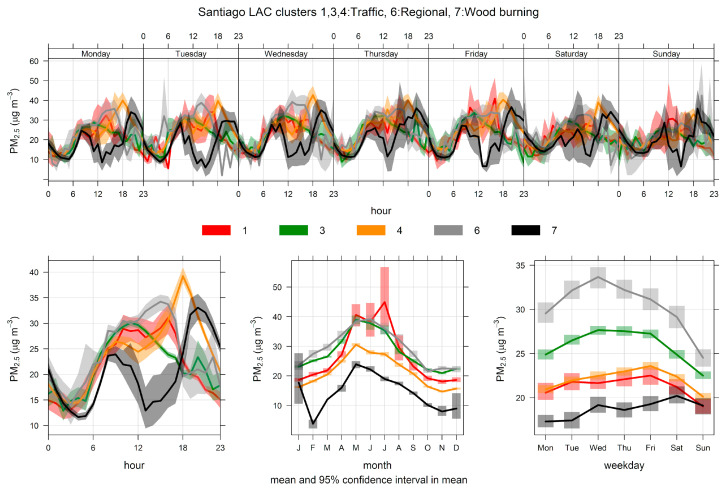
Time variation plot for the cluster analysis source apportionment results for ambient PM_2.5_ at Santiago, 2004/01–2012/06, for an eight-cluster solution. For clarity’s sake, only the five major clusters are included.

**Figure 15 ijerph-17-08455-f015:**
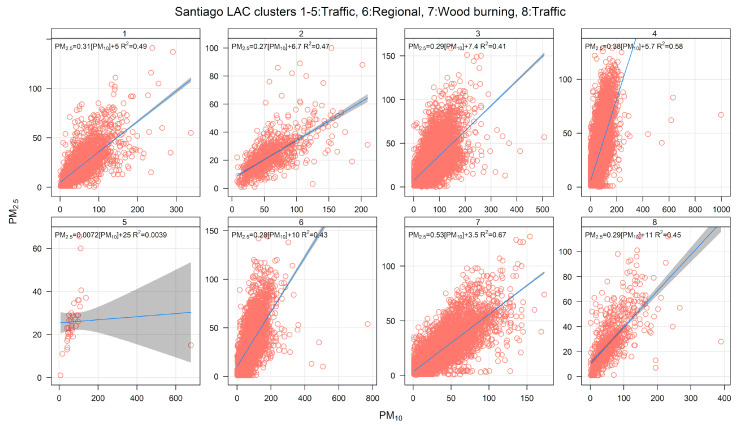
PM_2.5_–PM_10_ scatter plot by cluster, for an eight-cluster solution for Santiago.

**Table 1 ijerph-17-08455-t001:** Description of ambient monitoring sites analyzed.

City	Monitor	URL for Data Access	Period Analyzed
Calama	Centro	https://sinca.mma.gob.cl/index.php/estacion/index/key/236	10/2012–08/2020
Temuco	LE	https://sinca.mma.gob.cl/index.php/estacion/index/key/901	01/2009–08/2020
Santiago	LAC	https://sinca.mma.gob.cl/index.php/estacion/index/key/D13	01/2004–06/2012 ^1^

^1^ For the Santiago, Las Condes (LAC) site, meteorology is only available from 01/2004 through 06/2012.

**Table 2 ijerph-17-08455-t002:** Comparison for cluster analysis (CA) and receptor model (RM) source apportionment estimates (±standard error) for Temuco, July–September 2014 ^1^ in (µg/m^3^).

Period	RWB ^2^ (CA)	RWB (RM)	Traffic (CA)	Traffic (RM)
Week 1	35.3 ± 4.1	42.7 ± 5.9	0.9 ± 0.5	2.2 ± 0.2
Week 2	64.3 ± 7.1	75.9 ± 7.1	0.5 ± 0.5	2.4 ± 0.5
Week 3	15.2 ± 2.8	18.6 ± 5.0	3.2 ± 1.0	1.4 ± 0.1
Week 4	47.9 ± 5.6	52.6 ± 6.3	0.0 ± 0.0	2.3 ± 0.2
Week 5	52.9 ± 6.6	56.9 ± 6.7	3.7 ± 1.1	2.6 ± 0.2
Week 6	43.7 ± 4.4	42.8 ± 6.1	2.7 ± 1.0	2.5 ± 0.2
Week 7	23.1 ± 4.2	23.5 ± 5.0	5.1 ± 1.1	1.5 ± 0.1
Week 8	57.6 ± 9.0	48.6 ± 7.8	1.1 ± 0.3	2.8 ± 0.2
Average	42.5	45.2	2.2	2.2

^1^ Supplementary material in Reference [60] provides further details and data. ^2^ Residential wood burning sources.

**Table 3 ijerph-17-08455-t003:** Monthly source contributions to ambient PM_2.5_ by cluster for Santiago, year 2004 (µg/m^3^).

Month	C1	C2	C3	C4	C5	C6	C7	C8
1	2.1	0.3	10.4	5.8	0.0	0.9	0.0	0.1
2	2.7	0.6	11.4	9.1	0.0	1.8	0.0	0.0
3	1.8	0.3	11.0	10.7	0.0	2.4	0.0	0.2
4	1.7	0.2	5.8	13.3	0.02	3.7	0.7	0.5
5	0.4	0.1	5.7	17.4	0.0	6.6	6.5	0.4
6	0.1	0.0	4.0	16.7	0.0	5.9	4.5	1.3
7	0.0	0.0	3.1	16.0	0.0	4.7	3.9	0.9
8	0.2	0.0	4.7	9.2	0.0	3.2	6.8	0.4
9	0.4	0.0	6.3	10.8	0.0	3.1	2.1	0.4
10	1.0	0.1	6.3	8.0	0.0	2.8	0.6	0.3
11	1.5	0.3	7.7	6.0	0.0	3.2	0.1	0.4
12	2.4	0.5	10.4	5.7	0.0	1.5	0.0	0.3

**Table 4 ijerph-17-08455-t004:** Ambient PM_2.5_ source apportionment (±standard error), Santiago, May–August 2004 (µg/m^3^).

Month	Traffic (CA)	Traffic (RM)	RWB (CA)	RWB (RM)	Regional (CA)	Regional (RM)
May	25.2 ± 2.5	14.7 ± 1.2	7.4 ± 2.4	15.1 ± 1.6	6.3 ± 1.2	17.7 ± 2.2
June	20.8 ± 1.9	17.6 ± 1.7	6.4 ± 2.1	13.8 ± 1.2	7.9 ± 1.3	11.6 ± 1.1
July	21.6 ± 2.1	16.8 ± 1.5	4.0 ± 1.6	12.7 ± 1.3	6.7 ± 1.7	12.7 ± 1.7
August	14.8 ± 1.8	9.3 ± 0.8	8.1 ± 2.3	7.3 ± 0.7	4.0 ± 1.1	12.3 ± 1.3

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
