# Peer review of "Combining Cluster Analysis of Air Pollution and Meteorological Data with Receptor Model Results for Ambient PM2.5 and PM10"

_ijerph, 2020, doi:10.3390/ijerph17228455_

Round 1

Reviewer 1 Report

Summary

This study proposes a method to cluster air pollution (PM10 and PM2.5) and meteorological measurements (wind speed and temperature) in order to identify the major sources of the pollutants. The method is implemented using hourly averaged ambient air quality measurements and the polarCluster function in the Openair package in R. The method was demonstrated in three areas in Chile: the metropolis of Santiago, Calama in the Atacama Desert, and Temuco where residential wood burning is prevalent in the winter. Sources are linked with specific clusters based on seven rules. The cluster analysis results are compared to receptor modelling based on chemical speciation of ambient PM2.5 samples for a 4-month period. It is found that the cluster analysis method over-estimates the contribution of PM2.5 from traffic, but under-estimates the contribution of PM2.5 from residential wood burning and regional sources. The limitations of the cluster analysis method are discussed.

I really enjoyed reading this paper. It addresses a persist problem in air quality management: how to determine the relative contribution of different sources to ambient pollution levels on a year-round basis, with usually very limited chemical speciation measurements available. It is very clearly written, and shows a thorough knowledge of source characteristics, how meteorology affects dispersion of pollution and chemistry. Three distinct study areas are well selected. These results will contribute to a valuable dataset which can be used to inform policy and programmes which aim to alleviate health risks of air pollution.

Major Issues

I have identified no major shortcomings in this article.

Minor Issues

Please address the following:

  1. There are several minor grammatical errors, mainly in the prepositions used.

  1. In Figures 1-4, it would be useful if the clusters could be named based on the identified sources, and not just the cluster number.

  1. In Table 1, please indicate the units of the values.

Reviewer 2 Report

INTRODUCTION

      The contextualization was written in a very didactic form, exploiting definitions of several approaches to the atmospheric dispersion problem. Perhaps, for a scientific text, it would be more interesting to highlight the main outcomes of other studies about the application of Receptor Models (RM) and Cluster Analysis (CA) for other cities around the world. So that we, as readers, have a better idea of ​​the accuracy level of these techniques and what are the main difficulties and uncertainties involved. The inclusion of other studies from the literature will also guide the results presented in the following chapters, where there is no comparison with other cities, except for previous studies for Chile.

      Regarding the dispersion models, it is not true that they need complex information such as atmospheric modeling. There are several simple models based on analytical procedures which require just a few meteorological parameters to produce results. Possibly, in this case, place the dispersion models under discussion could be inappropriate, once the proposal of RM is also not unanimous within the scientific environment.

      Many important statements were presented within the manuscript without citations, which does not fit well into an introductory chapter. Some terms employed seem out of context usually used in the literature, such as: 'numerical meteorological modeling'. In this case, the most used term is ‘atmospheric modeling’. Others need conceptual revision or to be replaced. For example, line 47: “Meteorology also plays a role in driving the above-mentioned chemical and physical processes, adding seasonality to ambient PM2.5 at different time scales.”. Here, the term "meteorology" seems inappropriate. Perhaps, the use of "atmospheric conditions" would be more suitable for this context.

      The objectives were presented as a description of the activities. Therefore, it is not very clear what the general objective is or what question the authors propose to answer with this study.

QUESTION    In your opinion, which atmospheric process plays the main role in a conceptual model of air pollutant dispersion? Atmospheric chemistry or 'meteorology'?

QUESTION    Besides your analysis handles concentration time-series up to 16 years long, is it enough to classify this study as climatological? Further, the three sites analysed really represents three "widely different climate conditions" (see line 98)?

MATERIALS AND METHODS

      The computational methodology is very superficial and is not sufficient for a reproduction of the results afterwards. This part is the most critical part of the manuscript and will require the most effort. There is no introduction, description, and details of the Receptor Model CMB8.2, as well as the parameters employed. It was used only for Temuco analysis, which is contrary to one of the objectives of this study, i.e., to evaluate two different tools (RM and CA) to estimate the source apportionment of Particulate Material. A suggestion for this issue is to exclude RM from the main objectives and present it as an alternative tool for Temuco's analysis.

      The constant calling for figures and tables presented in the supplementary material made reading the manuscript a finicky task. The manuscript reading may not be prevented by the absence of material inside the body text. Table S1 and Figure S1, for example, are fundamental to section 2.2 and must be included in the manuscript. Supplementary material cannot be used as a toolbox in where authors place the whole package of results so that later reviewers can answer what results should be included in the body of the text.

      Typically, population information should be referenced. Thus, for each one of the three sites analysed, the citation is needed.

QUESTION    Does Section 2.3 present results produced by the authors in this study or does it present conclusions obtained from the literature review? If it comes from the literature, it should be cited. As presented, seems to be a conclusion obtained by authors about results not presented yet (until this section).

QUESTION    Why choose the receptor model Chemical Mass Balance (CMB8.2)? What equations are used? What parameters are needed? Is it better than the Positive Matrix Factorization (PMF)?

RESULTS

      This chapter is the best in the manuscript. Very interesting content with direct and coherent analysis, and well-structured presentation.

      Citation is needed. Also, corrections in figure and table captions.

      Once again, the excessive calling for the supplementary material muddled the outcomes discussion, where some important conclusions were drawn based on figures and tables not included in the manuscript. As a suggestion, try to summarize the results as a text, without calling the figures or indicating something specific in them. If this is not possible, consider adding the figure in the manuscript.

      There are unexplored figures in the manuscript which can be excluded to open space for some important results presented in the supplementary material.

      Finally, I recommend adding the polar clusters graphs presented only in the supplementary material and much explored in the manuscript: 5 Clusters - Calama; 3 Clusters - Temuco; and 8 Clusters - Santiago.

QUESTION    In Figure 2, the MP10 weekday concentration chart is enough to classify mobile sources as the main component of the three presented clusters (see lines 232-234)? Apparently, only Sunday shows a 10% reduction in comparison to the working days, approximately. A small amount to claim that traffic sources are primarily responsible for these concentration levels. This, without questioning the size of the vehicular fleet in Calama.

QUESTION    The same question could be applied to the analysis of Figure 3 (see lines 282-284). But for Temuco, there is no evidence of mobile sources influencing that could be mentioned only looking for Figure 3.

QUESTION    In which evidences the statement presented in lines 340-342 are based? Considering a city within a valley, does the anabatic wind act by removing or bringing pollutants to Santiago?

QUESTION    Why the cluster analysis for Santiago didn't consider the same concentration dataset used by Jorquera and Barraza (1999)?

DISCUSSION

      This chapter presents the same comments of the previous chapter (Results) and was written using a Conclusions chapter formula. Only the final part contains additional information (lines 438-459) and can be easily embedded in the following chapter (Conclusions).

Round 2

Reviewer 2 Report

First of all, I would like to congratulate the precise changes made in the manuscript. Although necessary, these changes elevate the study to the level that it really should be. This work is very relevant for large cities in South America, where the lack of environmental data limits the air quality studies and the application of various analysis techniques used for other large cities worldwide. This further worth your work, which sought alternative methods to enable this type of analysis.